# BrainM³: A Multi-Task Learning Framework Based on A Multi-Level Mixture-of-Experts for Cross-Disease and Cross-Domain Dementia Diagnosis

## Abstract

Accurate differential diagnosis of dementia subtypes is crucial due to their distinct clinical trajectories and treatment responses. However, rare subtypes such as Lewy Body Dementia (LBD) suffer from data scarcity, and domain shifts across institutions further hinder model generalization. To address these challenges, we propose **BrainM³**, a **M**ulti-task learning framework based on a **M**ulti-level **M**ixture-of-Experts (MoE) architecture for cross-domain and cross-disease **Brain** modeling. Our model jointly learns Alzheimer's disease (AD), mild cognitive impairment (MCI), and LBD diagnosis by disentangling disease-shared and specific brain connectivity features. At the domain level, a domain-aware Soft-MoE combined with adversarial training captures domain-invariant foundation brain representations, effectively mitigating scanner and cohort variability. At the task level, task-shared and task-specific Soft-MoEs enable mutual knowledge transfer and facilitate fine-grained pathological feature modeling. Experiments on multi-institutional datasets demonstrate that BrainM³ consistently outperforms baselines under data heterogeneity. Moreover, our model offers interpretable insights into disease-relevant brain networks, offering potential clinical utility. Our work highlights the promise of an accurate and interpretable model for robust dementia diagnosis in real-world, cross-institution settings. Our code will be published based on acceptance.

## 1 Introduction

Dementia poses a significant threat to human health and presents substantial clinical and socioeconomic challenges (Arvanitakis et al., 2019). Among the various subtypes, Alzheimer's disease (AD) is the most common, followed by Lewy body dementia (LBD) ranks second in prevalence (Outeiro et al., 2019; Erkkinen et al., 2018; Hugo & Ganguli, 2014; Orad & Shiner, 2022). Because each subtype follows a distinct clinical trajectory and responds differently to available therapies, accurate differential diagnosis is essential for guiding effective, personalized treatment strategies and for slowing disease progression (Xue et al., 2024).

In recent years, deep learning models have emerged as a powerful tool for automating the detection of various neurological disorders, with a variety of modeling strategies proposed for AD diagnosis. For instance, graph-based models (Song et al., 2019; Ma et al., 2020; Zhang et al., 2023; Zhou et al., 2024) capture inter-regional brain connections; transformer-based models (Chen et al., 2024; Zhang et al., 2024; 2025a) effectively model long-range dependencies; and more recently, state-space models (Cao et al., 2024; Chen et al., 2025; Ren et al., 2025) have been introduced as a computationally efficient alternative for learning global spatiotemporal representations. While AD has been extensively studied and benefits from large-scale public datasets, LBD remains underexplored due to data scarcity, which also limits the applicability of complex deep learning models. Despite recent efforts (Falaschetti et al., 2024; Wang et al., 2025; Zhang et al., 2025b) developing models for LBD diagnosis, a common limitation is that they are trained on data-scarce tasks, which may hinder generalization and robustness.

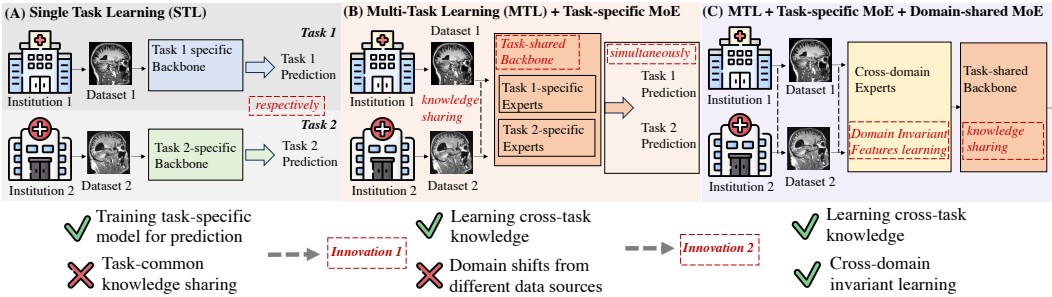

Figure 1: **Illustration of our innovations.** (A) Traditional single-task learning methods fail to capture task-shared knowledge. (B) *Our first innovation:* jointly learning multi-task in an unified MTL framework and introducing task-specific MoE modules, learning both task-shared and task-specific features to enhance the diagnostic performance on both tasks. (C) *Our second innovation:* incorporating a cross-domain MoE, facilitating the learning of domain-invariant representations and mitigating domain shift caused by heterogeneous data sources.

Given that AD and LBD share common pathological features yet also exhibit disease-specific traits (Yousaf et al., 2019), a natural intuition is that shared brain representations can be leveraged across tasks, enabling mutual knowledge transfer between data-rich and data-scarce settings. Multi-task learning (MTL) (See Fig. 1(B)) offers a promising solution, as it enables simultaneous learning of multiple related tasks to model cross-task relationships and take advantage of task-shared representations. (Zhang & Yang, 2021; Liu et al., 2024; Zhou et al., 2011), which can effectively improve model generalization and reduce data scarcity in data-scarce tasks. However, another major challenge arises from data heterogeneity. Brain imaging data collected from different sites often differ due to scanners, protocols, and patient populations, leading to significant domain shifts. Training a unified MTL model on such heterogeneous data potentially biases learning and causes negative knowledge transfer (Aoki et al., 2022). In real-world clinical diagnostic protocols, this heterogeneous-featured MTL setting is more realistic but remains underexplored.

To address both the challenges of cross-task knowledge transfer and cross-domain heterogeneity, we propose **BrainM**[3], a **M**TL framework based on a **M**ulti-level **M**ixture-of-Experts (MoE) architecture (See Fig. 1(C)). At the domain level, a domain-aware MoE module and a domain adversarial training strategy are introduced to learn domain-invariant brain foundational representations. At the task level, we incorporate task-shared and task-specific MoEs to disentangle disease-related representations from domain-dependent foundational representations, enabling synergistic learning across data-rich and data-scarce diagnosis tasks. This hierarchical design allows the model to adaptively capture disease-related pathological patterns in a collaborative learning way while maintaining robustness across varied cohorts and imaging protocols. We validate our proposed model by comparing it with several established baselines on disease prediction tasks from diverse institutions. Experimental results demonstrate that our BrainM[3] consistently outperforms baseline methods, highlighting the effectiveness. To the best of our knowledge, this is the first work to explore the heterogeneous-feature multi-task learning problem in brain disorder research. The main contributions of this work are summarized as follows:

**Domain-aware Adaptation:** We introduce a domain-level MoE and a domain adversarial training strategy into a unified MTL framework to eliminate domain shifts caused by heterogeneous data, learning domain-invariant yet informative brain foundational representations.

**Multi-task MoE Framework:** We propose a unified MTL framework that incorporates both task-shared and task-specific MoE modules to jointly learn from data-rich and data-scarce diagnosis tasks. A residual fusion strategy integrates shared and specific features, capturing both general and disease-specific patterns.

**Performance and Interpretability:** We conduct extensive experiments on data-rich and data-scarce datasets, demonstrating the model's superior performance and generalizability, particularly under data scarcity and domain variability conditions. We also offer analysis of explainable insights into the model's decision-making process and the disease-related pathologies.

## 2 PRELIMINARIES

### 2.1 PROBLEM DEFINITION

We begin by defining **heterogeneous-feature multi-task learning (MTL)** problem (Zhang & Yeung, 2011). Given a set of $m$ related but distinct tasks $\{T_i\}_{i=1}^m$, the goal is to jointly learn all tasks to improve performance on each individual task by leveraging shared knowledge. Unlike traditional homogeneous MTL where all tasks share the same input space, in heterogeneous-feature MTL, each task operates on a distinct feature space $\mathcal{X}_k$, i.e.,

$$y_k = T_k(B_k), \quad B_k \in \mathcal{X}_k, \quad \mathcal{X}_i \neq \mathcal{X}_j \text{ for } i \neq j \tag{1}$$

This setting mimics real-world scenarios where input distributions vary across tasks, yet each task still benefits from joint representation learning.

### 2.2 SOFT MIXTURE-OF-EXPERT(MOE)

MoE enables domain adaptation of heterogeneous data (Guo et al., 2018; Zhong et al., 2022; Jain et al., 2023; Wu et al., 2025b; Mi et al., 2025) and flexibly captures task-specific patterns (Wang et al., 2022; Fan et al., 2022; Chen et al., 2023; Zhu et al., 2024; Ding et al., 2025; Wu et al., 2025a) in a data-dependent routing. Soft MoE (Puigcerver et al., 2023) replaces the top-K hard selection mechanism in the sparse MoE (Mustafa et al., 2022) with a softmax-based token-to-expert assignment, allowing each input token to contribute to all experts in a weighted manner. Formally, given an input feature $x \in \mathbf{R}^{B \times N \times D}$, where $B$ is the batch size, $N$ is the number of brain sub-networks, and $D$ is the embedding dimension, a gating network computes logits $G(x) \in \mathbf{R}^{B \times N \times K}$ over $K$ experts. The softmax function then yields gating weights $\mathbf{w} = \text{softmax}(G(x)) \in \mathbf{R}^{B \times N \times K}$. Each expert $E_k$ processes the input $x$ independently to generate an output, and the final MoE product is a weighted combination of expert outputs: $\text{MoE}(x) = \sum_{k=1}^K w_k(x) \cdot E_k(x)$, where $w_k(x)$ denotes the weighting assigned to the expert $k$.

Our approach employs a shared Soft-MoE to learn domain-invariant foundational brain features, while task-shared and task-specific Soft-MoEs are introduced to capture both common and unique pathological patterns for each task, enabling multi-task knowledge sharing across diverse diseases and heterogeneous clinical populations.

### 2.3 BRAIN SUB-NETWORK REPRESENTATION

Brain structural connectivity (SC) patterns reflect disease-related alterations and offer informative representation of whole-brain network organization (Farooq et al., 2019; Škoch et al., 2022; Yeh et al., 2021; Zhang et al., 2021). To encode SC in a clinically meaningful and interpretable way, we introduce a brain-inspired token representation. Specifically, for a given subject, we applied standard imaging preprocessing (Zhang et al., 2022), including eddy current correction, fiber tracking, and registration of T1-weighted images to DTI space. Cortical segmentation was performed to parcellate the brain into 148 regions of interest (ROIs) based on the Destrieux Atlas (Destrieux et al., 2010). Pairwise connectivity strength between ROIs was computed based on the number of reconstructed white matter fibers, resulting in a symmetric matrix $X \in \mathbf{R}^{148 \times 148}$, where each entry $X_{ij}$ denotes the connectivity strength between ROI $i$ and $j$. Here, we define each column of the SC matrix as a ***brain sub-network***, capturing the connectivity fingerprint of a single ROI. This design preserves anatomical topology and enables region-wise expert selection within the Soft-MoE module.

## 3 METHOD

As illustrated in Fig. 2, our goal is to jointly model multi-task diagnosis from diverse institutions by leveraging shared and specific brain SC patterns. To this end, we propose BrainM$^3$, a framework built on a shared backbone that consists of three key components: (1) a domain-shared Soft-MoE module to learn domain-invariant foundation brain representations (See Fig. 2 (B-I)), (2) task-specific and task-shared Soft-MoE modules to capture both common and unique pathologies for each diagnosis task (See Fig. 2 (B-II)), and (3) a domain adversarial component to further enforce cross-domain generalization.

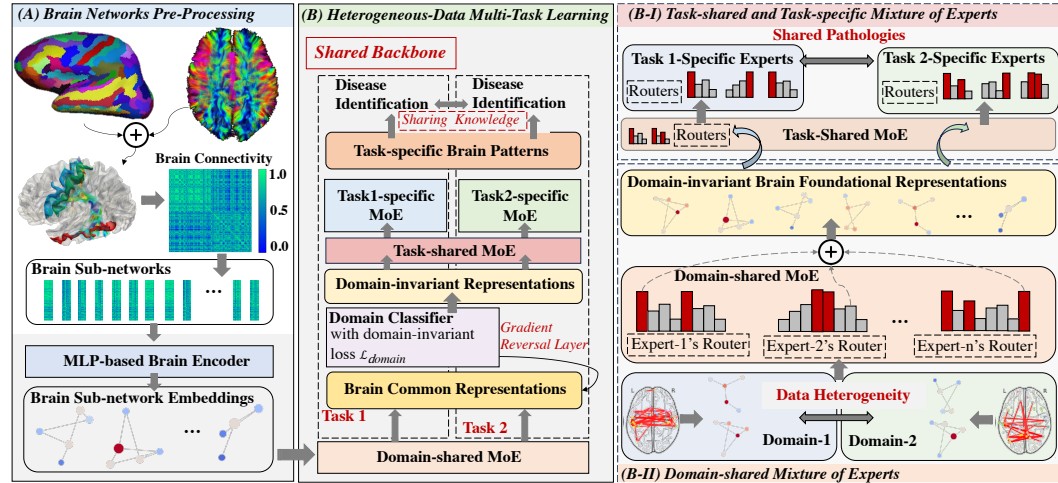

Figure 2: **Overview of our proposed framework**, which performs joint optimization across tasks under a unified shared backbone, including (A) brain networks processing and encoding, and (B) heterogeneous-feature MTL learning via hierarchical Soft-MoE.

## 3.1 DOMAIN-SHARED FEATURE LEARNING VIA SOFT-MOE

To extract domain-invariant and task-free foundation features from brain SC, we introduce a domain-shared soft-MoE module, which serves as the core of the shared backbone and is jointly optimized across tasks. Specifically, as shown in Fig. 2(A), let $X \in \mathrm{R}^{R \times R}$ denote the SC matrix of a subject, where $R$ is the number of brain regions. Each column vector $x_j \in \mathrm{R}^R$ reflects the connectivity profile of brain region $j$, capturing its relationship with all other regions. Each sub-network $x_j$ is first projected through a token-wise MLP encoder to yield an embedded representation:

$$h_j = \mathrm{MLP}_{\mathrm{enc}}(x_j) \quad j = 1, \dots, R \tag{2}$$

This produces an encoded sequence $H = [h_1, h_2, \dots, h_R] \in \mathrm{R}^{R \times D}$, where $D$ is the embedding dimension. To further obtain compact global brain network representations, we apply attention pooling over the sub-network sequence. Specifically, a learnable query vector $q \in \mathrm{R}^{1 \times D}$ attends to all tokens using multi-head attention:

$$z_{\mathrm{global}} = \mathrm{MultiHeadAttn}(q, H, H) \tag{3}$$

where the query attends to the sequence $H$ to produce a weighted global summary $z_{\mathrm{global}} \in \mathrm{R}^D$. This attention-based mechanism allows the model to focus on the most informative brain regions.

Subsequently, the encoded brain sub-networks are passed through a shared Soft-MoE layer consisting of $K$ experts $\{E_k(\cdot)\}_{k=1}^K$, each implemented as an independent MLP. A gating network computes a soft assignment weight $\alpha_{j,k} \in [0, 1]$ for each pair of brain sub-network and expert:

$$\alpha_{j,k} = \frac{\exp(g_k(h_j))}{\sum_{k'=1}^K \exp(g_{k'}(h_j))} \tag{4}$$

The output of the Soft-MoE is a weighted sum of expert outputs:

$$\hat{h}_j = \sum_{k=1}^K \alpha_{j,k} \cdot E_k(h_j) \quad j = 1, \dots, R \tag{5}$$

This results in a refined token representation $\hat{H} = [\hat{h}_1, \dots, \hat{h}_R] \in \mathrm{R}^{R \times D}$ that captures domain-robust brain structural features.

## 3.2 TASK-SHARED AND TASK-SPECIFIC FEATURE LEARNING

To capture both shared and specific disease patterns, we introduce a dual-branch feature specialization design comprising task-shared and task-specific Soft-MoEs, which operate sequentially on domain-shared brain representations.

Let $\hat{H} = [\hat{h}_1, \ldots, \hat{h}_R] \in \mathrm{R}^{R \times D}$ be the output of the domain-shared MoE, where each $\hat{h}_j \in \mathrm{R}^D$ is the embedding of brain sub-network $j$. We first feed this sequence into a task-shared MoE module, which is shared across tasks and designed to extract generalizable pathological representations:

$$s_j = \sum_{k=1}^{K_s} \alpha_{j,k}^{(s)} \cdot E_k^{(s)}(\hat{h}_j) \quad j = 1, \ldots, R \tag{6}$$

where $\alpha_{j,k}^{(s)}$ are the gating weights for the shared experts, and $E_k^{(s)}(\cdot)$ denotes the $k$-th task-shared expert.

The output $s_j \in \mathrm{R}^D$ is further processed by task-specific MoEs for each task $t$. Each task-specific MoE has its own set of experts and gating functions:

$$z_j^{(t)} = \sum_{k=1}^{K_t} \alpha_{j,k}^{(t)} \cdot E_k^{(t)}(s_j) \quad j = 1, \ldots, R \tag{7}$$

where $\alpha_{j,k}^{(t)}$ are the task-specific gating weights and $E_k^{(t)}(\cdot)$ is the $k$-th expert for task $t$.

To effectively integrate shared and task-specific information, we apply a residual fusion mechanism by summing the outputs of the task-shared and task-specific MoEs:

$$\tilde{z}_j^{(t)} = s_j + z_j^{(t)} \quad j = 1, \ldots, R \tag{8}$$

The fused sequence $\tilde{Z}^{(t)} = [\tilde{z}_1^{(t)}, \ldots, \tilde{z}_R^{(t)}] \in \mathrm{R}^{R \times D}$ is then aggregated into a global representation via attention pooling:

$$z_{\mathrm{global}}^{(t)} = \mathrm{MultiHeadAttn}(q, \tilde{Z}^{(t)}, \tilde{Z}^{(t)}) \tag{9}$$

followed by a task-specific classification head:

$$\hat{y}^{(t)} = \mathrm{Classifier}^{(t)}(z_{\mathrm{global}}^{(t)}) \tag{10}$$

where $\hat{y}^{(t)} \in \mathrm{R}^{C_t}$ is the prediction for task $t$ with $C_t$ classes.

### 3.3 DOMAIN ADVERSARIAL LEARNING

Despite the use of a shared feature extractor, subtle domain-specific variations may persist. To further promote domain-invariant representation learning, we introduce a domain adversarial training strategy based on a gradient reversal mechanism. Specifically, Let $H_{\mathrm{shared}} \in \mathrm{R}^{B \times R \times D}$ denote the output token sequence from the domain-shared Soft-MoE, where $B$ is the batch size, $R$ is the number of brain regions, and $D$ is the embedding dimension. To obtain global representations, we apply attention pooling over the sequence:

$$z_{\mathrm{shared}} = \mathrm{MultiHeadAttn}(q, H_{\mathrm{shared}}, H_{\mathrm{shared}}) \tag{11}$$

A domain classifier $f_{\mathrm{dom}}(\cdot)$ is trained to predict the domain label $d \in \{0, 1\}$. During training, we apply a gradient reversal layer (GRL) (Ganin et al., 2016) before the domain classifier:

$$\hat{d} = f_{\mathrm{dom}}(\mathrm{GRL}(z^{\mathrm{shared}})) \tag{12}$$

where GRL multiplies the incoming gradients by a negative scalar $\lambda$, encouraging the feature extractor to produce domain-invariant embeddings. The domain classification loss is computed as:

$$\mathcal{L}_{\mathrm{domain}} = \frac{1}{B} \sum_{i=1}^{B} \mathrm{CE}(\hat{d}_i, d_i) \tag{13}$$

where $\mathrm{CE}(\cdot, \cdot)$ is the cross-entropy loss between the predicted label $\hat{d}_i$ and the ground truth $d_i$ of domain $i$.

This adversarial training paradigm formulates a minimax game, where in the domain classifier $f_{\mathrm{dom}}(\cdot)$ is optimized to distinguish between data domains, while the feature encoder is simultaneously trained to produce domain-invariant representations that confuse the classifier.

### 3.4 TRAINING OBJECTIVES

The proposed framework is trained end-to-end by jointly optimizing classification and domain adversarial losses. Let $\mathcal{L}_{\mathrm{dr}}$ and $\mathcal{L}_{\mathrm{ds}}$ denote the cross-entropy classification losses for the data-rich and data-scarce tasks, respectively. $\mathcal{L}_{\mathrm{domain}}$ is the domain adversarial loss defined in section 3.3. The overall training objective is:

$$\mathcal{L}_{\mathrm{total}} = \mathcal{L}_{\mathrm{dr}} + \mathcal{L}_{\mathrm{ds}} + \lambda \cdot \mathcal{L}_{\mathrm{domain}} \tag{14}$$

where $\lambda$ is a hyperparameter for domain regularization.

During training, mini-batches from both data sources are alternately fed into the model. The shared encoder and domain-shared Soft-MoE are optimized jointly across tasks, while the task-specific MoEs and classifiers are trained independently per task. The domain classifier is trained adversarially via a gradient reversal layer to encourage the extraction of domain-invariant features. As shown in Fig. 2(B), this joint optimization strategy enables the model to leverage shared knowledge across tasks while maintaining task-specific discriminability and robustness to domain shift.

## 4 EXPERIMENTS

### 4.1 EXPERIMENTAL SETTINGS

**Datasets** In this study, we evaluated our proposed BrainM$^3$ model on two datasets: a public dataset from the Alzheimer's Disease Neuroimaging Initiative (ADNI) (Jack Jr et al., 2008), which offers a relatively large sample size, and an in-house dataset from an anonymized institution, characterized by limited data availability. The ADNI dataset is used for distinguishing normal controls (NC) from individuals with mild cognitive impairment (MCI), a prodromal stage of AD. After quality control, 418 subjects were included (301 NC and 117 MCI). The in-house dataset contains 147 subjects (23 NC, 77 LBD, and 47 AD). Data preprocessing followed the procedures described in section 2.3 *Brain Sub-network Representation*. We used 80% of the data for training and 20% for testing on each dataset. Additional preprocessing details and ADNI subject demographics are provided in the supplementary material.

**Implementation Details** In our method, we set the embedding dimension of each brain sub-network to 64. The MoE modules consist of 8 domain-shared experts, 4 task-shared experts, and 4 task-specific experts for each diagnosis task. The balancing hyperparameter $\lambda$ for the $\mathcal{L}_{\mathrm{domain}}$ is set to 1. The BrainM$^3$ is trained with a batch size of 16 for 128 epochs using the Adam optimizer with a learning rate of $1 \times 10^{-4}$. All experiments are conducted on a workstation equipped with an NVIDIA RTX 6000 GPU.

### 4.2 COMPARISON WITH BASELINES

We evaluate the proposed BrainM$^3$ on multi-type dementia identification tasks using both the ADNI dataset and a private dataset. The performance is compared against several baseline methods, including two traditional machine learning models: Support Vector Machine (SVM) and XGBoost, two CNN/GNN-based approaches: BrainNetCNN (Kawahara et al., 2017) and FBNETGEN (Kan et al., 2022a), and two Transformer-based methods: VanillaTF (Kan et al., 2022b) and BrainNetTF (Kan et al., 2022b). All baseline models are trained on *single diagnosis task* separately, with hyperparameters adopted from their original papers. In the row "BrainM$^3$ (Single-task)" of Table 1, we ablate both the domain-shared and task-shared modules and *train each single task independently* to enables a fair comparison.

*From a single-task perspective*, CNN/GNN-based approaches consistently outperform traditional machine learning models by capturing complex brain topological patterns, and transformer-based methods further improve performance on both tasks by learning long-range global dependencies. Our model outperforms all baselines on the private dataset across all evaluation metrics, demonstrating its advantage in data-scarce scenarios. Compared to baseline methods, BrainM$^3$ integrates both local sub-network modeling and flexible expert routing, enabling data-adaptive representation learning that is more robust to limited learning samples. In contrast, baseline deep learning models may suffer from overfitting due to their high complexity.

Table 1: Performance comparison of different baselines on the ANDI and Private datasets. The best results within single-task learning are highlighted in underline, the overall best results across all methods are highlighted with **bold**.

| Methods | ANDI (data-rich) | | | | Private (data-rare) | | | |
|---|---|---|---|---|---|---|---|---|
| | ACC | AUROC | SEN | SPE | ACC | AUROC | SEN | SPE |
| SVM | 60.00 | 75.01 | 50.19 | 57.50 | 63.33 | 75.15 | 53.89 | 55.56 |
| XGBoost | 63.33 | 75.32 | 63.89 | 63.53 | 70.00 | 75.47 | 55.97 | 75.59 |
| BrainNetCNN | 64.29 | 73.13 | 63.33 | 76.67 | 73.33 | 78.52 | 58.52 | 81.00 |
| FBNETGNN | 70.24 | 62.30 | 63.75 | 75.00 | 74.78 | 81.84 | 60.14 | 82.54 |
| VanillaTF | 71.43 | 73.78 | 66.77 | 73.33 | 76.33 | 82.50 | 62.50 | 83.92 |
| BrainNetTF | 73.62 | 75.07 | 67.17 | 77.00 | 75.97 | 81.97 | 60.88 | 84.13 |
| BrainM$^3$ (Single-task) | 70.16 | 79.81 | 67.50 | 73.33 | 76.67 | 84.54 | 66.04 | 82.46 |
| BrainM$^3$ (Multi-task) | **84.38** | **85.69** | **69.34** | **88.68** | **80.00** | **88.13** | **72.12** | **87.30** |

Notably, when ***both tasks are trained jointly***, performance improves significantly across the board, particularly on the ADNI dataset, where accuracy increases by 14.19%. Moreover, both tasks achieve approximately 5% improvements in AUROC, as illustrated in the row "BrainM$^3$ (Multi-task)" of Table 1. These highlight the effectiveness of cross-task representation learning. Through mutual learning between data-rich and data-scarce neurological-related tasks, our BrainM$^3$ improves performance by positive knowledge transfer and offers a promising direction for maximizing data use in real-world data-rare medical settings.

To further understand the model's behavior, we analyze the confusion matrix for the CN vs AD vs LBD diagnosis task. As shown in Fig. 4. the model correctly identifies most LBD and AD samples, but struggles with CN. This is potentially due to limited CN cases, which may lead to insufficient pattern learning. In contrast, it performs well on both AD and LBD diagnosis, which have relatively more samples. Notably, our BrainM$^3$ achieves clear separation between AD and LBD, a clinically challenging task due to their overlapping symptoms. This demonstrates that our BrainM$^3$ is capable of learning fine-grained disease-specific features, even in challenging, data-rare scenarios.

### 4.3 INTERPRETABILITY

Beyond predictive accuracy, an ideal diagnostic model should also offer interpretability by uncovering disease-relevant brain patterns and decision-making process, which is essential for clinical trust. As shown in Fig. 3, we visualize the top-5 discriminative brain sub-networks for each diagnosis task based on expert activation scores from the task-specific MoE module, highlighting task-specific brain SC patterns that contribute most significantly to the model's predictions (See Fig. 3(A-B)). In addition, we identify the top-5 task-shared brain sub-networks by analyzing expert activation scores in the task-shared MoE, providing evidence of common pathological substrates across different dementia types (See Fig. 3(C)).

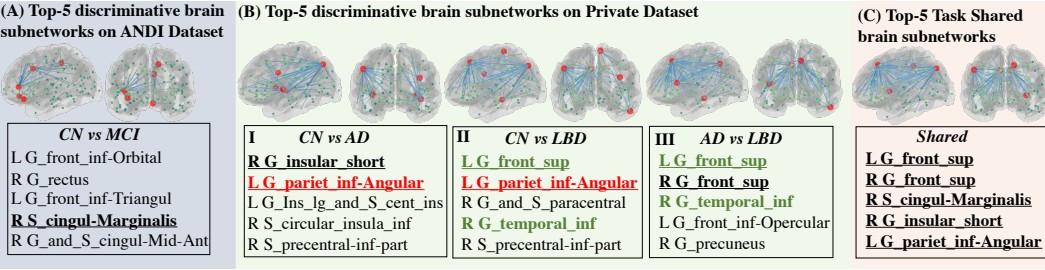

Figure 3: Top-5 discriminative brain sub-networks identified for each task, and the Top-5 most activated task-shared sub-networks. Shared sub-networks across diseases are highlighted in **bold** with different colors, while across task-specific and task-shared sets are additionally underlined.

**Explainable Result on ADNI Dataset** Figure 3 (A) illustrates the top-5 discriminative brain SC sub-networks distinguishing between CN and MCI, which are high aligned with previous studies. For instance, both the *L G_front_inf-Orbital* and the *R G_rectus* have been reported to show significantly reduced gray matter volume in individuals with MCI compared to CN (Han et al., 2012; Xie et al., 2015). The *L G_fron_inf-Triangul*, has been widely implicated in neurodegenerative diseases, especially those impacting language-related functions (Whitwell et al., 2015; Reyes et al., 2018; Mandelli et al., 2016). The *R G_and_S_cingul-Mid-Ant* belongs to the anterior cingulate cortex, a region frequently affected by structural, functional, and metabolic alterations in neurodegenerative conditions (Jones et al., 2006; Yuan et al., 2022). Additionally, the *R S_cingul-Marginalis* is located near a key hub of the default mode network (DMN), where AD-related pathological changes such as amyloid-$\beta$ deposition and cortical atrophy often occur (Leech & Sharp, 2014). Notably, this region is also identified as a task-shared sub-network (see Fig. 3 (C)).

**Explainable Result on Private Dataset** Figure 3 (B) highlights the top-5 discriminative SC brain sub-networks that differentiate CN, AD, and LBD. As expected, our model effectively captures disease-related brain alterations. Frist, the *R G_insular_short*, located in the anterior insula, is a critical hub linking sensory perception, emotional processing and autonomic regulation (Uddin et al., 2017). This region is known to undergo significant structural atrophy in AD (Fathy et al., 2020) and is successfully identified in our method (see Fig. 3 (B-I)), and is also recognized as a task-shared sub-network (see Fig. 3 (C)). Moreover, the *L G_pariet_inf-Angular*, functions as a multimodal convergence zone and is a key hub within the brain's DMN(Wagner & Rusconi, 2023; Wang et al., 2019). In both AD and LBD, this region shows pronounced glucose metabolism reduce (Lim et al., 2009). Our model successfully identifies this sub-network as discriminative between both CN vs. AD and CN vs. LBD (see Fig. 3 (B-I, B-II)), and it is also recognized as a task-shared sub-network (see Fig. 3 (C)).

Furthermore, the *L G_front_sup* serves as a critical hub in multiple brain networks including DMN (Li et al., 2013). Both AD and LBD exhibit cortical atrophy and reduced glucose metabolism in this region, with LBD showing greater posterior involvement and a distinct metabolic pattern (Yousaf et al., 2019; Mistur et al., 2009). Similarly, the *R G_temporal_inf*, a region essential for high-level visual recognition and multimodal integration (Onitsuka et al., 2004), shows structural and functional abnormalities in LBD, while its relatively preserved metabolism compared to AD highlights its diagnostic relevance (Mak et al., 2014; Shivamurthy et al., 2015; Barber et al., 2000). Our model identifies these sub-network as discriminative in both CN vs LBD and AD vs LBD comparisons (see Fig. 3 (B-II, B-III)), and the *L G_front_sup* is identified as a task-shared sub-network (see Fig. 3 (C)). More importantly, the *R G_front_sup* is identified as both a discriminative sub-network between AD and LBD (see Fig. 3, (B-III)) and a task-shared sub-network (see Fig. 3 (C)). Prior studies have reported significant frontal lobe atrophy, reduced glucose metabolism, and disrupted functional connectivity in this region among individuals with AD, whereas these alterations are generally milder in LBD (Yousaf et al., 2019; Valdés Hernández et al., 2018; Tang et al., 2021; Roquet et al., 2016). Such distinctions are clinically important for differentiating between AD and LBD. Our method effectively captures these sub-network, highlighting the clinical interpretability of our model in capturing disease-specific brain alterations.

### 4.4 ABLATION STUDIES

**Number of Experts** We categorize experts into three types: *domain-shared*, *task-shared*, and *task-specific*. In this ablation study, we vary the number and composition of experts to examine (1) the necessity of each expert type and (2) the effect of expert count in each MoE block on overall model performance. Experiments are conducted based on the BrainM$^3$ and the results are summarized in Table 2. Overall, models incorporating domain-shared experts consistently outperform those with the same total number of experts but without domain-shared ones, with 8 domain-shared experts yielding the best performance. Moreover, setting the number of experts in any individual MoE block to zero leads to a performance drop, underscoring the essential role of each MoE block in identifying brain diseases.

**Effect of $\mathcal{L}_{\text{domain}}$** To investigate the impact of the domain adversarial loss $\mathcal{L}_{\text{domain}}$, we conduct a sensitivity analysis by varying its balancing weight $\lambda$ in the overall objective. Fig. 5 presents the classification performance on both datasets under different $\lambda$ values. As $\lambda$ increases from 0 to 1, performance on both datasets improves significantly, with the best result achieved at $\lambda = 1$. When

$\lambda$ exceeds 1, the performance on the ADNI dataset drops slightly, indicating that an overly strong domain alignment may suppress task-discriminative features. Additionally, we ablate the GRL in the domain classifier and the result shows that when it is removed, performance on both tasks drops. These results validate the effectiveness of the proposed domain adversarial mechanism in mitigating domain shift.

Table 2: Ablation Study on Expert Number and Structure. Here, $m$ denotes the total number of experts, $m_d$ denotes the number of domain-shared experts, $m_s$ denotes the task-shared experts, and $m_t$ denotes the task-specific experts. The best results under the same $m$ are shown in underlined, while the overall best results across all settings are **bold**.

| $m$ | $m_d$ | $m_s$ | $m_t$ | ANDI (data-rich) | | | | Private (data-rare) | | | |
|---|---|---|---|---|---|---|---|---|---|---|---|
| | | | | ACC | AUROC | SEN | SPE | ACC | AUROC | SEN | SPE |
| 2 | 2 | 0 | 0 | 77.37 | 80.22 | 67.44 | 82.35 | 70.09 | 85.87 | 63.54 | 83.15 |
| 2 | 0 | 2 | 0 | 75.00 | 67.43 | 57.14 | 82.15 | 70.00 | 82.87 | 61.02 | 80.16 |
| 2 | 0 | 0 | 2 | 75.10 | 68.57 | 68.12 | 82.33 | 70.00 | 83.97 | 65.60 | 82.25 |
| 4 | 4 | 0 | 0 | 78.12 | 79.33 | 56.10 | 88.51 | 73.33 | 82.02 | 64.72 | 84.38 |
| 4 | 0 | 4 | 0 | 75.10 | 67.51 | 54.29 | 82.19 | 70.94 | 86.98 | 64.19 | 82.19 |
| 4 | 0 | 0 | 4 | 77.34 | 69.22 | 63.44 | 84.23 | 72.67 | **89.32** | 63.54 | 83.46 |
| 4 | 2 | 2 | 0 | 78.91 | 85.12 | 69.31 | 85.24 | 71.79 | 87.26 | 60.70 | 82.52 |
| 4 | 0 | 2 | 2 | 76.56 | 80.44 | 69.75 | 87.87 | 70.94 | 86.98 | 69.19 | 84.19 |
| 8 | 8 | 0 | 0 | 82.03 | 85.61 | 67.26 | 83.33 | 73.50 | 86.29 | 63.76 | 83.88 |
| 8 | 0 | 8 | 0 | 75.00 | 66.86 | 54.29 | 83.15 | 66.77 | 81.97 | 54.35 | 78.03 |
| 8 | 0 | 0 | 8 | 76.07 | 69.19 | 68.01 | 85.48 | 70.00 | 81.97 | 58.06 | 80.16 |
| 8 | 4 | 4 | 0 | 81.25 | 85.21 | 63.54 | 83.20 | 74.36 | 87.52 | 61.05 | 83.53 |
| 8 | 4 | 0 | 4 | 82.81 | 83.78 | 65.71 | 83.77 | 75.21 | 87.91 | 65.84 | 84.82 |
| 8 | 4 | 2 | 2 | 82.03 | **86.57** | 69.19 | 83.19 | 76.67 | 87.05 | 71.39 | 85.97 |
| 16 | 8 | 0 | 8 | 83.59 | 85.58 | 67.06 | 86.81 | 74.36 | 88.84 | 62.34 | 83.87 |
| 16 | 0 | 8 | 8 | 79.69 | 80.41 | 66.13 | 85.65 | 71.70 | 87.06 | 67.69 | 82.78 |
| 16 | 8 | 8 | 0 | 82.87 | 83.43 | 67.14 | 86.77 | 73.33 | 81.16 | 64.72 | 83.59 |
| 16 | 8 | 4 | 4 | **84.38** | 85.69 | **69.34** | **88.68** | **80.00** | 88.13 | **72.12** | **87.30** |

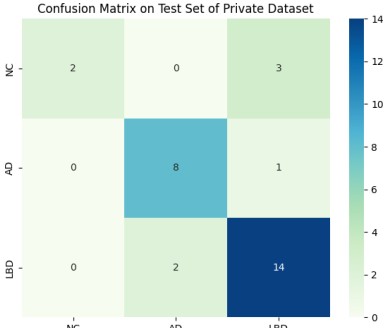

Figure 4: Confusion matrix for three-class diagnosis task.

Figure 5: The impact of the hyperparameter $\lambda$ on both tasks.

## 5 CONCLUSION

We propose BrainM³, a novel MTL framework with a Multi-level Mixture-of-Experts architecture, designed to address both cross-domain heterogeneity and cross-task knowledge transfer in brain disorder diagnosis. The domain-shared MoE effectively mitigates domain shift, while task-specific and task-shared MoEs benefits knowledge transfer between date-rich and date-rare neurological tasks. Beyond superior performance, our model offers interpretable insights into disease-relevant brain sub-networks, providing potential clinical relevance. To our best knowledge, our work represents the first exploration of heterogeneous MTL in brain disorder research, establishing a foundation for robust, cross-institutional dementia diagnosis.

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

# A APPENDIX

**Subject Demographics and Preprocessing details of ADNI**

The imaging protocol for each subject included structural MRI (T1-weighted) and diffusion tensor imaging (DTI). T1-weighted images were acquired with a field of view (FOV) of 240 mm × 256 mm × 208 mm, isotropic voxel size of 1.0 mm, and repetition time (TR) of 2.3 s. DTI data were obtained using a b-value of 1000 s/mm², 54 gradient directions, FOV of 232 mm × 232 mm × 160 mm, isotropic voxel size of 2.0 mm, TR of 7.2 s, and echo time (TE) of 56 ms. Preprocessing steps included skull stripping for both modalities, followed by registration of T1 images to DTI space using FSL. T1 images underwent tissue segmentation via FreeSurfer, with regions of interest (ROIs) defined according to the Destrieux Atlas. DTI preprocessing involved eddy current correction using FSL, followed by fiber tracking reconstruction using MedINRIA.

Demographic details of the ADNI subjects are presented in Table 3, including sample size, sex distribution, and age (mean ± standard deviation) for both NC and MCI.

Table 3: Demographic information of subjects from ADNI

|  | Mean ± standard deviation | |
| --- | --- | --- |
|  | NC | MCI |
| Sample size | 301 | 117 |
| Male/female | 118/183 | 74/43 |
| Male age (years) | 71.37 ± 5.92 | 73.21 ± 6.86 |
| Female age (years) | 70.25 ± 5.91 | 70.17 ± 7.32 |

**Brain Structural Connectivity Visualization Across Datasets**

To qualitatively assess the structural differences across datasets, we visualize the brain structural connectivity matrices of three randomly selected subjects from each diagnostic group in both the ADNI and private datasets (See Figure 6).

In the ADNI dataset, both NC and MCI groups display structured and dense connectivity patterns, with symmetric topologies and moderate connection strengths. In contrast, the connectivity matrices from the private dataset (NC, AD, and LBD groups) are sparser. The distinct patterns highlight substantial distributional shifts between the two datasets, reinforcing the presence of cross-domain heterogeneity in brain structural connectivity. This observation motivates the integration of domain-invariant representation learning in our framework to enhance model robustness and generalizability.

**The use of large language models (LLMs)**

LLMs were used solely to improve the clarity and fluency of the language in this manuscript. All research ideas, methodology, experiments, analyses, and conclusions were conceived and carried out by the authors.

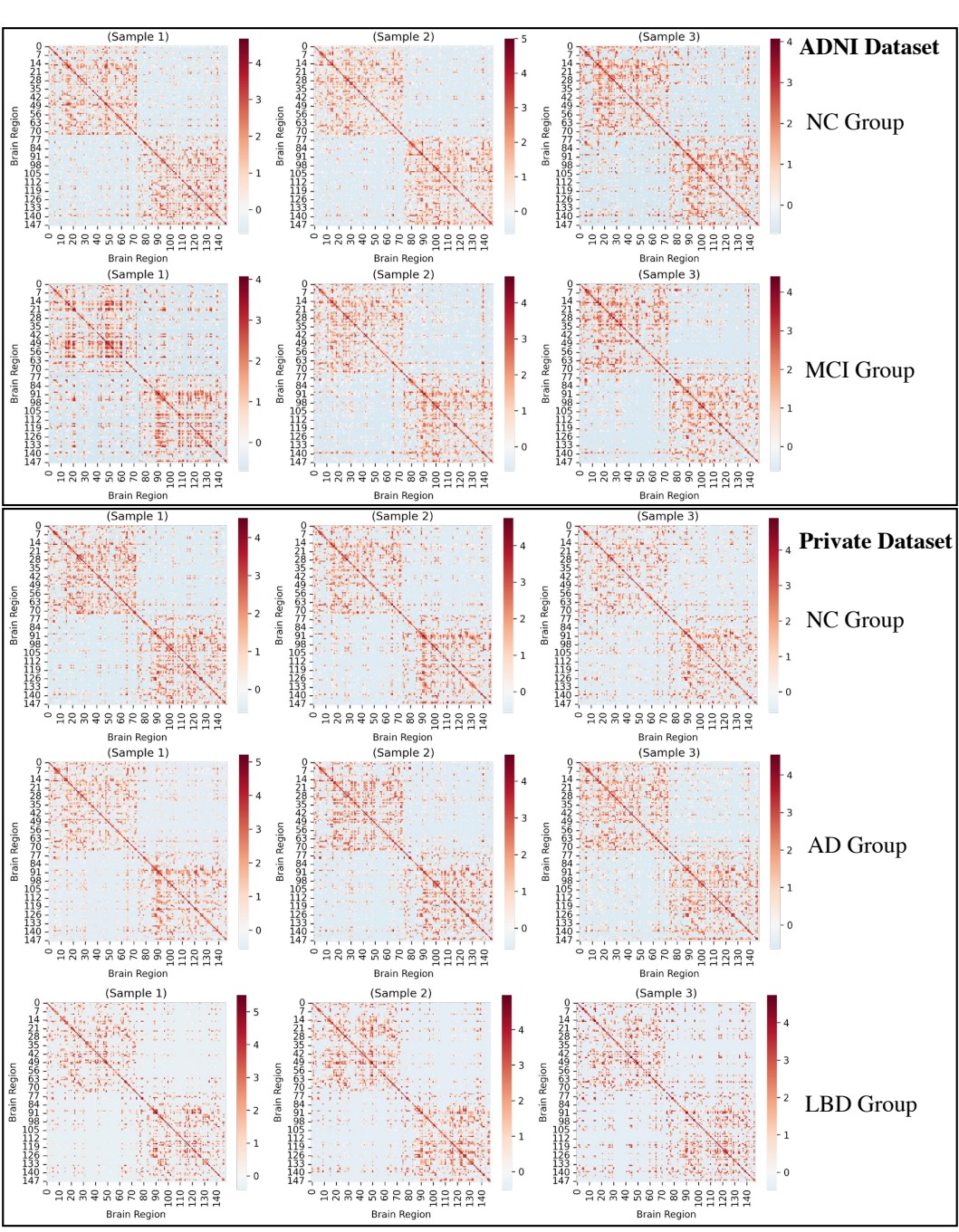

Figure 6: Brain structural connectivity matrices from ADNI and private datasets show distinct patterns across groups, revealing a clear domain shift in connectivity distributions.

