# OpenReview forum: "$\text{BrainM}^3$: A Multi-Task Learning Framework Based on A Multi-Level Mixture- of-Experts for Cross-Disease and Cross-Domain Dementia Diagnosis"
_ICLR.cc/2026/Conference — Submitted to ICLR 2026_

### Official Review · Reviewer_UJ9P · 2025-10-17

**Soundness:** 2
**Presentation:** 3
**Contribution:** 2
**Rating:** 4
**Confidence:** 3

**Summary:**

This paper proposes a model called BrainM3, designed to simultaneously handle the diagnosis of AD, MCI, and LBD, while also accounting for data variability across hospitals and scanners. The model's core design is a hierarchical MoE architecture. Domain-level MoEs first learn universal features that are robust across institutions. Task-shared and task-specific MoEs then use them to enable data-rich tasks like AD to benefit data-limited tasks like LBD. The model also utilizes adversarial training to mitigate domain shift. Experiments show that the multi-task version of BrainM3 performs significantly better than single-task versions, particularly in scenarios with limited data. The authors also analyze the brain regions targeted by the model, which are consistent with established medical literature, demonstrating a degree of interpretability.

**Strengths:**

The paper's starting point is very realistic. Rather than simply playing with the model in the “clean world” of ADNI, it truly considers the common problem of “model obsolescence if the hospital changes.” The model's structural design is logical: domain --> shared task --> task-specific. This layered arrangement is well-reasoned and not simply a stacking of modules. In experimental comparisons, multi-task performance significantly outperformed single-task performance, especially on the LBD task, demonstrating that the model truly transfers knowledge, not simply sharing parameters. The authors also specifically analyze the brain regions activated by the model. This interpretability is a plus in the medical field and is much more impressive than pure accuracy.

**Weaknesses:**

1. Although the paper title emphasizes “heterogeneous-feature multi-task learning,” the three tasks actually share the same input format (the structural connection matrix), but the inputs come from different sources. Strictly speaking, these are not considered truly “heterogeneous features.”

2. The baselines used for model comparison are almost all single-task learning, with no comparison to simpler multi-task shared models (e.g., a shared encoder connected directly to three heads). This makes it difficult to determine whether MoE is truly effective, or whether simply training with a shared backbone will suffice. Furthermore, the baseline methods are significantly outdated.

3. The private dataset is relatively small, and the current experiments are more like "domain-aware fine-tuning," so the conclusions about "cross-domain generalization" still have room for improvement.

4. While the model architecture incorporates MoE, the paper does not demonstrate its gating behavior, such as whether different experts truly specialize in different tasks or data sources. Without visualization, one might suspect that MoE simply adds parameters rather than creating a division of labor.

**Questions:**

1. Have simpler multi-task baselines been developed? For example, using only a shared encoder and task head, without a MoE, how much difference is there in performance? If the performance is similar, then the value of the MoE needs to be redefined.

2. Can cross-domain capabilities be tested more aggressively? For example, if the model's performance is tested directly without training on private data, what will happen? This would further validate the domain-invariant claim.

3. Can you show the activation maps or routing weights of the experts? We could see if some experts clearly favor LBD or ADNI data, which would make the role of the MoE more intuitive.

4. Can the term “heterogeneous features” be more precise? Currently, the input structures of the three tasks are not truly different. In the future, are you considering adding fMRI or PET to achieve truly multimodal heterogeneous input?

---

> ### Author Response · Authors · 2025-11-26
>
> We thank the reviewer for the valuable feedback. Per the rebuttal guidelines, we address the main concerns and outline plans for the revised version following reviewers’ suggestions.
>
> Q1. Simpler multi-task baselines:
> We appreciate the reviewer’s question. To the best of our knowledge, this is the first work to explore multitask learning on brain structural connectome data; therefore, there are no established MTL baselines in this domain. For fairness, we included single-task learning results to benchmark against existing methods. We agree that evaluating simpler multi-task variants is valuable. In the revision, we will add results for a shared encoder with task-specific heads.
>
> Q2. Expert activation/routing visualization:
> Thank you for the insightful suggestion. We will include routing-weight and expert-activation visualizations in the revised manuscript.
>
> Q3. Definition of “heterogeneous features”:
> We appreciate the clarification. In our work, “heterogeneous” refers to distributional heterogeneity, arising from differences in disease populations (AD vs. LBD) and cross-site acquisition characteristics, rather than modality-level heterogeneity. Although all tasks use SC matrices, their underlying statistical structures differ substantially across tasks and domains. We will refine the terminology accordingly.
>
> Q4. Future extension to multimodal inputs:
> Thank you for the suggestion. Yes, our future work aims to extend our framework to multimodal heterogeneous-feature learning, incorporating additional modalities such as fMRI or PET to further help disease learning.

---

> > ### Comment · Reviewer_UJ9P · 2025-11-27
> >
> > I appreciate the authors' rebuttal and clarifications. While their responses address the questions generally, they do not fully resolve my main concerns about technical validation and empirical evidence. As a result, my overall assessment still stands.

---

### Official Review · Reviewer_Acp6 · 2025-10-28

**Soundness:** 2
**Presentation:** 2
**Contribution:** 2
**Rating:** 4
**Confidence:** 4

**Summary:**

This paper proposes BrainM³, a multi-task learning framework based on a multi-level Mixture-of-Experts (MoE) architecture for dementia subtype diagnosis under real-world challenges: data scarcity and cross-institutional domain shift. The method jointly models Alzheimer’s disease (AD), mild cognitive impairment (MCI), and Lewy body dementia (LBD) by disentangling domain-invariant foundational representations and task-shared/specific features. The authors validate their approach on a public dataset (ADNI) and an in-house private dataset, demonstrating consistent performance gains over strong baselines and providing interpretable insights into disease-relevant brain networks.

**Strengths:**

1.The use of Soft-MoE with attention-based pooling enables flexible, region-aware routing, which is particularly suitable for brain connectivity data.

2.The gradient reversal layer (GRL) combined with domain-shared MoE enables model to learn domain-invariant features without sacrificing task discriminability.

**Weaknesses:**

1. The ANDI dataset used in this study includes only Alzheimer's disease–related tasks, indicating that the multi-task experiments are conducted within the same domain rather than across different domains.

2. The design of task-specific and task-shared MoE architectures is not new; it has been extensively explored in prior studies such as [A, B, C].

  [A]Dynamic modeling of patients, modalities and tasks via multi-modal multi-task mixture of experts. ICLR 2025.

  [B]TaskExpert: Dynamically Assembling Multi-Task Representations with Memorial Mixture-of-Experts. ICCV 2023.

  [C]Exploring Text-enhanced Mixture-of-Experts for Semi-supervised Medical Image Segmentation with Composite Data. MICCAI 2025.

3. The comparison is insufficient and somewhat outdated, as the most recent competing method cited was published in 2022.

4. The discussion is insufficient. There are alternative approaches to achieving domain-invariant representation learning, with one of the most classical being the Fourier-based transformation, which has been widely applied in domain generalization tasks. The authors are encouraged to discuss or compare their method with such alternatives.

5. The interpretability section still requires further refinement, as higher performance metrics naturally lead to more precise sub-network discriminability. However, this alone does not demonstrate the interpretability of the proposed method.

6. Section 2.1 introduces heterogeneous-feature multi-task learning (MTL), yet the remainder of the paper never clarifies how “heterogeneous” MTL differs from “homogeneous” MTL in terms of generalization bounds or optimization difficulty, nor does it explain how the proposed multi-level MoE mitigates the effects induced by these differences.

**Questions:**

1. The overall contribution appears incremental. Please clarify which specific models are incorporated within the proposed MoE.

2. There are many alternative designs for MoE architectures, such as pMoE (Patch-level Routing in Mixture-of-Experts is Provably Sample-efficient for Convolutional Neural Networks). Please consider discussing or comparing your approach with these alternatives.

3. The current comparisons are not comprehensive enough and could benefit from including more recent or diverse baselines.

---

> ### Author Response · Authors · 2025-11-26
>
> We thank the reviewer for the valuable feedback. Per the rebuttal guidelines, we address the main concerns following reviewers’ suggestions.
>
> Q1. Definition of Multi-Task Setting:
> ANDI dataset indeed only includes AD-related tasks, but our work introduces the LBD dataset, a different dementia subtype, yielding a heterogeneous task pair. Thus, the multi-task setting spans different disease categories.
>
> Q2. Clarification on “Heterogeneous-feature MTL”:
> In our setting, heterogeneity arises from task-specific brain connectome distributions and differences in cross-site acquisition. The proposed multi-level MoE mitigates these through domain-shared experts and adversarial alignment.
>
> Q3. Alternative domain-invariant approaches (e.g., Fourier-based DG):
> We appreciate this suggestion. Fourier-based domain generalization is effective for image-space perturbations, but brain connectome matrices represent graph-structured anatomical connectivity, for which Fourier perturbations are not directly applicable.
>
> Q4: Alternative MoE architectures (e.g., pMoE) should be discussed:
> We appreciate the suggestion. pMoE is designed for grid-structured spatial patches in convolutional architectures, whereas brain connectomes use biologically defined region-level tokens. Consequently, patch-level routing is not compatible with connectivity-based representations.
>
> Q5. Novelty of the Proposed Approach:
> We agree that MoE-based MTL has been explored previously. Our contribution lies not in introducing MoE itself, but in presenting the first multi-level MoE framework specifically designed for heterogeneous brain connectome learning. To our knowledge, this is the first work applying such a hierarchical MoE structure in brain disorder research.
>
> Q6. Comparison is insufficient or outdated:
> Thank you for the suggestion. We note that brain-connectome-based learning remains relatively underexplored, and available strong baselines are limited, but we will expand the comparison accordingly.

---

### Official Review · Reviewer_kmzm · 2025-11-02

**Soundness:** 3
**Presentation:** 3
**Contribution:** 3
**Rating:** 4
**Confidence:** 4

**Summary:**

The paper introduces BrainM3, a multi-task learning (MTL) framework built upon a multi-level mixture-of-experts (MoE) architecture for cross-domain and cross-disease dementia diagnosis. It aims to address two core challenges: Data scarcity in rare dementia subtypes like Lewy Body Dementia (LBD). Domain heterogeneity arising from different imaging sites and scanners. BrainM3 integrates three levels of feature specialization: Domain-shared Soft-MoE for domain-invariant foundational representations. Task-shared Soft-MoE for common pathological representations. Task-specific Soft-MoE for disease-unique characteristics. The framework also uses adversarial training via a Gradient Reversal Layer to enforce domain invariance.

**Strengths:**

1. Novel hierarchical architecture: The combination of domain-level, task-level, and expert-level specialization is elegant and technically coherent. Incorporation of soft MoE routing avoids the instability of sparse gating and enhances flexibility.
2. Addresses practical challenges: Cross-domain heterogeneity and data scarcity are real-world problems in neuroimaging, and the proposed multi-level structure effectively models them.
3, Comprehensive evaluation: Comparison against diverse baselines (traditional ML, CNN, GNN, Transformer). Includes ablation studies on expert number and domain-adversarial loss, confirming design choices.

**Weaknesses:**

1. Limited dataset diversity: Only two datasets are used. The cross-domain setting, though well motivated, remains narrow in scope. Absence of external validation or additional domains limits claims of “generalization”.
2. Potential overfitting to small datasets: The private dataset includes only 147 subjects; despite MoE and regularization, results may reflect overfitting rather than robust generalization.
3. Interpretability remains qualitative: While visualizations are compelling, quantitative interpretability metrics like stability of identified regions and overlap with known biomarkers are not reported.

**Questions:**

1. Scalability: How would BrainM3 scale to more than two diagnostic tasks or more than two data domains?
2. Generalization and robustness: Did you test model on any unseen external dataset to confirm cross-site generalization beyond ADNI and your private dataset?

---

> ### Author Response · Authors · 2025-11-26
>
> We thank the reviewer for the valuable feedback. Per the rebuttal guidelines, we address the main concerns following reviewers’ suggestions.
>
> Q1. Limited dataset diversity and potential overfitting on a small dataset
>
> We appreciate the reviewer’s concern. While only two datasets are used, AD and LBD are the most clinically relevant dementia subtypes in neuroimaging research, and publicly available LBD datasets are incredibly scarce. Our private LBD cohort is, to the best of our knowledge, among the largest reported and reflects the real-world data scarcity that our method is designed to address. This limitation arises from the broader neuroimaging landscape rather than from our approach. Accordingly, our goal is precisely to maximize the utility of heterogeneous cross-site data under such limited availability.
>
> Q2. Scalability to more diagnostic tasks or data domains
>
> Our framework is intentionally modular. The backbone remains unchanged when new tasks or domains are added, only additional task- or domain-specific Soft-MoEs are appended. Because Soft-MoE routing scales linearly with the number of experts, the overall computational cost grows sub-linearly, enabling BrainM³ to scale efficiently to larger and more heterogeneous clinical cohorts without architectural changes.

---

### Official Review · Reviewer_j8Lx · 2025-11-03

**Soundness:** 2
**Presentation:** 3
**Contribution:** 2
**Rating:** 4
**Confidence:** 4

**Summary:**

The paper proposes BrainM3, a multi-task learning (MTL) framework that leverages a hierarchical Soft Mixture-of-Experts (MoE) architecture to address dementia diagnosis across diseases (Alzheimer's disease [AD], mild cognitive impairment [MCI], and Lewy Body Dementia [LBD]) and domains (different institutions with heterogeneous data). The model utilizes domain-shared MoE with adversarial training to learn domain-invariant representations of brain structural connectivity (SC) from diffusion tensor imaging (DTI). It then employs task-shared and task-specific MoEs to capture common and unique pathological patterns, enabling knowledge transfer from data-rich (AD/MCI) to data-scarce (LBD) tasks. Experiments on ADNI and an in-house dataset claim superior performance over baselines, with interpretability insights into brain networks.

**Strengths:**

- The work tackles a timely and clinically significant challenge: differential dementia diagnosis under data scarcity (e.g., for LBD) and domain shifts (e.g., scanner variability). This aligns with real-world medical imaging scenarios, where datasets are often heterogeneous and imbalanced. The focus on interpretability (e.g., via expert routing and attention pooling) could provide value beyond accuracy, such as identifying disease-relevant brain sub-networks.
- The hierarchical MoE design is an acceptable contribution to MTL in heterogeneous settings. Combining domain-shared MoE with adversarial training (via gradient reversal layer) for invariance, followed by task-shared/specific MoEs with residual fusion, is a smart way to disentangle features. This extends Soft-MoE (Puigcerver et al., 2023) to medical imaging, potentially advancing domain generalization in neuroimaging. The brain sub-network tokenization (based on Destrieux Atlas) preserves anatomical structure, making the model interpretable and biologically grounded.
- The use of multi-institutional datasets (ADNI for data-rich AD/MCI classification, in-house for data-scarce NC/AD/LBD) simulates cross-domain challenges. Comparing against diverse baselines (SVM, XGBoost, CNN/GNNs like BrainNetCNN, and Transformers like BrainNetTF) is appropriate, and the single-task ablation of BrainM3 enables a fair assessment of MTL benefits.

**Weaknesses:**

- The 80/20 train/test split is mentioned, but no cross-validation (e.g., 5-fold) is described, which is critical for small datasets (ADNI: 418 subjects; in-house: 147), which raises concerns about overfitting.
- Ablation studies (e.g., removing domain adversarial or task-shared MoE) are not done with the required details: does domain invariance truly mitigate shifts, or is it marginal? The hyperparameter sensitivity is not addressed (e.g., λ, number of experts K).
- Using SC from DTI is justified, but why not utilize multimodal input (e.g., fMRI + DTI) for richer features? The tokenization (148 ROIs as sub-networks) assumes symmetry in the matrix, but no handling of noise/artifacts in fiber tracking is discussed—e.g., how robust is it to preprocessing variations?
- Gating is softmax-based, but no load balancing loss (common in MoE to prevent expert collapse) is mentioned. With small datasets, experts might under-specialize; evidence suggests that the embedding dim D=64 and expert counts (8 domains, 4 shared/specific) seem arbitrary (the authors should provide a proper ablation needed).
- λ=1 is fixed in adversarial training; how sensitive is it? The domain classifier is simple (MLP), but no architecture details are provided. Does it overfit on small batches?
- The manuscript provides tests on only two datasets/domains. No external validation (e.g., UK Biobank or other dementia cohorts) is presented.
- The interpretability claims (e.g., "insights into disease-relevant networks") are vague without examples, for example, which ROIs are routed to which experts for LBD vs. AD?
- The paper acknowledges data scarcity but downplays ethical issues.
- No mention of computational complexity or accessibility for low-resource labs.

**Questions:**

- What specific strategies were used to address class imbalance during training?
- If oversampling was applied, was it performed at the subject level or within mini-batches? Was it combined with data augmentation (e.g., random fiber tract perturbations or graph perturbations)?
- Were these strategies applied uniformly across both tasks, or tailored per task/dataset?
- Please report per-class performance metrics (precision, recall, F1-score) in addition to overall accuracy/AUC to demonstrate that minority class performance is not sacrificed.
- The paper claims that BrainM3 offers “interpretable insights into disease-relevant brain networks” and that expert routing enables “fine-grained pathological feature modeling.” However, no visualizations or quantitative analyses of gating behavior, attention maps, or expert specialization are provided in the main paper.
Please provide:
- Gating weight heatmaps or t-SNE/UMAP projections of expert assignments per brain sub-network (ROI) across NC, MCI, AD, and LBD subjects. Do certain experts consistently activate for specific ROIs?
- Attention pooling visualizations from Eq. (9): which brain regions receive high attention in task-specific vs. task-shared pathways? Are these anatomically plausible?
- Expert specialization analysis: for each task-specific expert, compute the average activation (gating weight) across disease groups. Is there statistically significant specialization (e.g., via ANOVA or permutation tests)?
- A case study showing how routing differs between a correctly vs. incorrectly classified LBD subject.
- Comparison with Grad-CAM or integrated gradients on baseline models (e.g., BrainNetTF) to quantify whether MoE improves the localization of known pathology.
- Hyperparameter search details, such as grid search for K, λ?
Several critical hyperparameters are fixed without justification:
- Domain-shared experts: 8
- Task-shared experts: 4
- Task-specific experts: 4 per task
- Domain adversarial weight: λ = 1
- Embedding dimension: D = 64
- No load-balancing loss for MoE
Please report the search space and method used for hyperparameter tuning.

---

> ### Author Response · Authors · 2025-11-26
>
> We thank the reviewer for the valuable feedback. Per the rebuttal guidelines, we address the main concerns and outline plans for the revised version following reviewers’ suggestions.
>
> Q1. Strategies for addressing class imbalance:
> We thank the reviewer for the question. No explicit imbalance-handling strategies were applied.
>
> Q2. Reporting per-class performance metrics
> The per-class diagnostic behavior is shown in the confusion matrix for the three-class task (Fig. 4), and discussed in the final paragraph of Section 4.2. Notably, the minority and clinically challenging class (LBD) is reliably identified, indicating that its performance is not sacrificed.
>
> Q3. Hyperparameter search details:
> We appreciate the reviewer’s attention to this point. Hyperparameter details are already provided:
> The numbers of domain-shared, task-shared, and task-specific experts are summarized in Table 2 (Ablation Study on Expert Number and Structure). The sensitivity of the domain-adversarial weight λ is analyzed in Figure 5. We will make these references more explicit in the revision.
>
> Q4. Why not multimodal inputs:
> We agree that multimodal integration is valuable. However, such multimodal acquisitions are not available for our LBD cohort, and multimodal LBD datasets are extremely rare in the broader neuroimaging landscape. This limitation reflects data availability rather than a limitation of our approach. Extending our framework to multimodal settings is an important direction for future work.
>
> Q5. Robustness to DTI noise and preprocessing variations:
> Our preprocessing pipeline includes log-normalization, removal of spurious fibers, and z-scoring, which collectively improve robustness to tractography variability. We will clarify these steps in the revised manuscript.
>
> Q6. Interpretability: gating and expert specialization:
> Thank you for the insightful suggestion. In the revision, we will add gating-weight heatmaps to show how expert routing differs across diagnostic groups, demonstrating meaningful expert specialization and enhancing interpretability.
>
> Q7. Lack of cross-validation:
> We appreciate the reviewer’s concern. In the revision, we will add 5-fold cross-validation and report the variance to demonstrate stability.
>
> Q8. Computational complexity:
> We will add a concise complexity analysis. Importantly, the proposed framework introduces minimal inference overhead, as MoE routing is lightweight and experts operate on compact ROI-level token embeddings.

---

### Meta-Review · Area_Chair_2P9M · 2026-01-06

**Summary:**

The paper proposes BrainM^3, a multi-level mixture-of-experts framework for multi-task dementia diagnosis across diseases (AD, MCI, LBD) and domains (sites/scanners). Reviewers broadly agree the problem setting is realistic and the architecture is reasonable. The disagreement is mostly about strength of empirical validation and novelty: whether the MoE complexity is justified given limited data, limited domains, and missing baselines/analyses, and whether the interpretability and generalization claims are sufficiently supported.

**Reviewer Concerns:**

1. Empirical validation / overfitting risk (small data, limited domains): Partially addressed. The rebuttal commits to adding cross-validation and clarifies data scarcity constraints, but no new evidence is shown. External validation remains unaddressed.
2. Missing or weak baselines (simpler MTL, newer methods): Partially addressed. Authors acknowledge the issue and promise to add shared-encoder MTL baselines and expand comparisons, but this is future work, not resolved in the current version.
3. Interpretability and MoE justification (expert specialization, gating behavior): Partially addressed. The rebuttal explains intent and promises visualizations and analyses, but reviewers’ concern that MoE may just add parameters is not fully resolved without results.

**Reviewer Scores:**

Reviewer j8Lx (4 > 4 likely): Many concerns deferred to revision; core validation gaps still stand.
Reviewer kmzm (4 > 4 likely): Rebuttal reasonably defends scope and scalability, but dataset limitation remains.
Reviewer Acp6 (4 > 4 likely): Novelty and comparison concerns are acknowledged but not convincingly resolved.
Reviewer UJ9P (4 > 4 likely): Reviewer explicitly states their assessment still stands after rebuttal.

---

### Decision · Program_Chairs · 2026-01-26

Reject